# The Oxygen and Glucose Deprivation of Immature Cells of the Nervous System Exerts Distinct Effects on Mitochondria, Mitophagy, and Autophagy, Depending on the Cells’ Differentiation Stage

**DOI:** 10.3390/brainsci13060910

**Published:** 2023-06-04

**Authors:** Denis Jagečić, Dražen Juraj Petrović, Iva Šimunić, Jasmina Isaković, Dinko Mitrečić

**Affiliations:** 1Laboratory for Stem Cells, Croatian Institute for Brain Research, University of Zagreb School of Medicine, Šalata 3, 10 000 Zagreb, Croatia; dpetrovic@genos.hr (D.J.P.); ji520@nyu.edu (J.I.); dinko.mitrecic@mef.hr (D.M.); 2Genos d.o.o., Laboratory for Glycobiology, 10 000 Zagreb, Croatia; 3Omnion Research International, 10 000 Zagreb, Croatia

**Keywords:** neural stem cells, oxygen–glucose deprivation, mitochondria, autophagy, mitophagy

## Abstract

Perinatal brain damage, one of the most common causes of lifelong impairment, is predominantly caused by a lack of oxygen and glucose during early development. These conditions, in turn, affect cells of the nervous tissue through various stages of their maturation. To quantify the influence of these factors on cell differentiation and mitochondrial parameters, we exposed neural cell precursors to oxygen and glucose deprivation (OGD) during three stages of their differentiation: day 1, day 7, and day 14 (D1, D7, and D14, respectively). The obtained results show that OGD slows down cellular differentiation and causes cell death. Regardless of the level of cell maturity, the overall area of the mitochondria, their length, and the branching of their filaments decreased uniformly when exposed to OGD-related stress. Moreover, the cells in all stages of differentiation exhibited an increase in ROS production, hyperpolarization of the mitochondrial membrane, and autophagy. Interestingly, day 7 was the only stage in which a significant increase in mitochondrial fission, along with measurable instances of mitophagy, were detected. Taken together, the results of this study suggest that, apart from common reactions to a sudden lack of oxygen and glucose, cells in specific stages of neural differentiation can also exhibit increased preferences for mitochondrial fission and mitophagy. Such findings could play a role in guiding the future development of novel therapeutic approaches targeting perinatal brain damage during specific stages of nervous system development.

## 1. Introduction

The development of the nervous system and its subsequent functionality boasts a sequential differentiation of cellular precursors, the most prominent of which are neural stem cells (NSCs). NSCs are multipotent stem cells that give rise to neurons, astrocytes, and oligodendrocytes [1]. Unlike the adult nervous system, in which the ratio of undifferentiated cells is negligible, the immature nervous system comprises cells in various developmental stages [2,3,4].

Brain diseases, including stroke and perinatal ischemia, pose a great burden on modern society and can result in lifelong disability [5]. From a pathophysiological perspective, the sudden lack of oxygen and glucose supply to the nervous system results in a systemic decrease in membrane potential, reduction in ATP production, and neuronal swelling [6]. In turn, the cellular metabolism is changing from aerobic to anaerobic, leading to pH imbalance and cell death [7,8]. Although the occurrence of necrosis and apoptosis has been well documented in cases of nervous tissue ischemia, there is also evidence that ischemic insult could lead to mitochondrial dysfunction, followed by autophagy and mitophagy. Autophagy is a highly conserved cellular process comprising the selection and degradation of damaged intracellular organelles, or proteins, within lysosomes. Even though it could be activated by many physiological stimuli, the process of basal autophagy is generally active under normal conditions and constitutes an important homeostatic mechanism for maintaining a healthy cellular environment. In addition, autophagy has also been recognized as one of the major mechanisms underlying perinatal brain damage [9,10]. Even though some researchers reported that promoting autophagy after brain ischemia is beneficial, others demonstrate the opposite to be true [11,12]. Thus, the role of autophagy in stroke remains controversial.

Mitophagy is a subtype of autophagy that involves the lysosome-dependent clearance of damaged mitochondria. This is particularly prominent in cases of ischemic injury of cells, which involves the impaired function of mitochondria, coupled with dysregulation of calcium homeostasis and the production of reactive oxygen species (ROS) [13]. Following such injuries, the success of subsequent improvement in mitochondrial metabolism can dictate the outcome regarding cell repair or cell death via necrosis, apoptosis, or autophagy [14]. Accordingly, it is not surprising that mitophagy is recognized as one of the processes that might become an important target in treating perinatal brain damage [15]. Even though some research studies suggest that increased mitophagy has beneficial effects in directing the recovery of the nervous tissue, it remains unclear how an increase or decrease in the rate of mitophagy would impact perinatal brain damage [16].

Furthermore, even though many publications reported the occurrence of autophagy and mitophagy in the nervous system during various stages of its development, including in the neonatal neurons, a direct comparison of cells throughout these stages of differentiation is lacking [17]. Therefore, the main goal of this work was to use our expertise in neural cell precursors and investigate the influence of oxygen and glucose deprivation on their early, middle, and late stages of differentiation. Specifically, our research focused on the differentiation and survival of immature cells of the nervous system, namely the quantification of the morphological and functional characteristics of their mitochondria upon exposure to the acute lack of oxygen and glucose, combined with a detailed overview of subsequent mitophagy and autophagy. As a result, we show that cells in various stages of maturity react differently to the lack of oxygen and glucose and that various parameters, such as the balance in the fusion/fission of mitochondria, change accordingly. Interestingly, we also note that neural cell precursors caught in the middle stage of differentiation (7 days) exhibit more significant changes in mitochondrial morphology. These findings are complemented by changes in several markers of mitophagy and autophagy. Hence, the results of this study suggest that neural cell precursors exhibit different and distinct reactions to the lack of oxygen and glucose throughout different stages of their differentiation. This finding might be important in choosing appropriate therapeutic approaches for the treatment of the immature nervous system affected by perinatal brain damage.

## 2. Materials and Methods

### 2.1. Isolation and Differentiation of Neural Stem Cells

NSCs were isolated from the telencephalic wall of 14.5-day-old C57/BL6 albino mice embryos. They were cultivated in a proliferation medium made from 1% N2 (Gibco, 17502-048, New York, NY, USA), 1% Pen/Strep (penicillin/streptomycin, 5000 U/mL, Gibco, 15070063), 2% B27 (Gibco, 17502), 20 ng/mL EGF (epidermal growth factor, Gibco, PMG8041), 10 ng/mL bFGF (fibroblast growth factor basic, Gibco, PMG0035), and 5 mM HEPES (Sigma-Aldrich, H0887, Burlington, MA, USA), where they formed structures called neurospheres. To initiate the process of their differentiation, the cells were cultivated in pretreated plates containing 50 μg/mL of poly-D-lysin (PDL, Sigma-Aldrich, P6407) and 10 μg/mL of laminin (Sigma-Aldrich, L2020). Herewith, the complete proliferation medium was replaced with the differentiation medium comprising 1% N2, 1% Pen/Strep, 1% FBS (fetal bovine serum, Gibco, 15070063), 2% B27+ (Gibco, A3582801), and 5 mM of HEPES. Half of the old medium was replaced with a new differentiation medium every 4 days.

### 2.2. Oxygen–Glucose Deprivation Model

To induce ischemic injury in 3 specific timepoints, we utilized the OGD (oxygen–glucose deprivation) method. Briefly, this included the replacement of the complete differentiation medium with Dulbecco’s no-glucose medium (Gibco, 11966025) on differentiation days 1 (D1), 7 (D7), and 14 (D14). After this, the cells were placed in a low-oxygen (1% O2) incubator for 24 h, after which the samples were taken. The samples from the control group were collected after the same period, albeit from the cells that were placed in a separate incubator and were grown under normal conditions.

### 2.3. Immunocytochemistry

The cells were grown on coverslips and fixed in 4% PFA for 10 min, followed by 3 washes with PBS. Permeabilization was performed in 0.2% Triton in PBS for 15 min, followed by another 3 washes with PBS. The cells were then blocked in filtered 3% goat serum in PBS at room temperature for 2 h. The primary antibodies were directly added onto the coverslips and incubated overnight at +4 °C. The following day, the cells were washed in PBS 3 times, after which a secondary antibody was added and incubated at room temperature for 1 h. Here, we used 1:1000 goat anti-mouse 488, 1:1000 goat anti-chicken 546, and 1:500 goat anti-rabbit 633. The cells were, once again, washed 3 times in PBS. The counterstain procedure was carried out using 1: 20,000 DAPI (4′,6-Diamidine-2′-phenylindole dihydrochloride, Roche, 10236276001) 

### 2.4. Western Blot

Protein lysates were prepared in a RIPA lysis buffer using ice and a 27 G needle with the addition of protease (Roche, 11836170001, Indianapolis, IN, USA) and phosphatase (Roche 4906837001) inhibitors. Quantification was performed using a detergent-compatible Bradford reagent (Thermo Scientific, 1863028, Waltham, MA, USA). The proteins were loaded into a 12% stain-free gel, and blocking was performed using 3% low-fat milk for 2 h. The membrane was incubated overnight at +4 °C with the following antibodies: anti-PINK1 (Abcam, ab23707, Waltham, MA, USA) 1:1000, anti-LC3 (Cell Signalling, #3868) 1:1000, anti-FUNDC1 (Novus Biologicals, NBP1-81063, Centennial, CO, USA) 1:2500, and anti-p62 1:15,000 (Abcam, ab109012). The membranes were then washed 3 times in TBST and incubated with secondary antibodies (1: 200,000) for 1 h. Subsequent detections were accomplished using the Femto SuperSignal chemiluminescent reagent (Thermo Scientific, 34095). Because of the low protein yield on D1, Western blot was only performed on D7 and D14 of differentiation. The results obtained with the Western blot were normalized to the total protein amount. All images of the Western blots can be found in the Appendix A.

### 2.5. Quantitative Polymerase Chain Reaction (qPCR)

To isolate RNA, the samples were first stored in an RLT buffer. The RNeasy kit (Qiagen, 74104, Hilden, Germany) was used for RNA isolation, and the concentration was measured using the Nanodrop device. Equal amounts of RNA (25 ng/µL) were later transcribed to cDNA using a high-capacity RNA to cDNA kit (Applied Biosystems, 4374966, Waltham, MA, USA). qPCR was performed by adding the same amount of the specific TaqMan assay and the sample of interest. mRNA was then analyzed using specific TaqMan probes for nestin, GFAP, MAP2, and FUNDC1. β-Actin was used as a housekeeping probe. Finally, the qPCR results were depicted as 2–∆∆Ct, which indicates the fold difference.

### 2.6. Lactate Dehydrogenase (LDH) Detection Kit

For the purposes of evaluating cellular damage, LDH-GloTM Cytotoxicity Assay (Promega, J2380, Madison, WI, USA) was used. Firstly, 5 µL of the medium was mixed with 95 µL of a storage buffer (200 mM Tris-HCl (pH 7.3), 10% Glycerol, 1% BSA). The samples were then combined with an LDH detection enzyme mix (detection enzyme and reductase) in a 1:1 ratio. The incubation time was 30 min, and the luminescence was detected with a 0.9 integration time. The amount of released LDH was proportional to the luminescence signal. The percentage of cytotoxicity was calculated according to the following formula: 100 × (cellular LDH release − medium background)/(maximum LDH release − medium background), where cellular LDH release denotes the level of LDH in specific conditions. On the other hand, the maximum LDH release denotes the amount of released LDH following a 15 min treatment with Triton X-100, which caused complete damage to the cells.

### 2.7. MitoSOX

In order to detect the mitochondrial oxygen radicals, we used a MitoSOX red detection kit (ThermoFisher, M36008, Waltham, MA, USA). Once it enters the live cells, it is oxidized by superoxide, resulting in the appearance of a fluorescent signal. Briefly, a 500 nM working concentration was directly added to the cells and incubated for 15 min. Cells were then washed 2× with preheated HBSS, after which the signal was detected at 580 nm. The oxygen radicals were observed using live-cell imaging with a glass bottom Petri dish and an Olympus FV3000 confocal microscope. Images were analyzed in Imaris 9.9 (Oxford Instruments, Oxfordshire, UK).

### 2.8. TMRE

To evaluate the mitochondrial membrane potential (MMP), we used a TMRE detection kit (Abcam, 113852) that is designed to quantify changes in the MMP of live cells. It consists of a positively charged dye that selectively binds to negatively charged active mitochondria. The dye was added to the cells at a final concentration of 200 nM for 20 min. The cells were then washed with preheated PBS twice, and a signal was detected at 575 nm. The mitochondrial membrane potential was, once again, observed using live-cell imaging.

### 2.9. LC3-GFP/Lysotracker Colocalization

As a way of analyzing colocalization, we used a Premo Autophagy Assay (ThermoFisher, P36235) and LysoTracker Deep Red (ThermoFisher, L12492). The Premo Autophagy sensor is based on non-replicative BacMaM technology for the stable transduction of LC3B-GFP chimera. The multiplicity of infection (MOI) was 20, and the incubation time was 24 h. LysoTracker Deep Red was used in a final concentration of 50 nM. The kit was incubated for 30 min, after which the medium was replaced with a new preheated medium. Images were analyzed in Imaris 9.9 (Oxford Instruments).

### 2.10. Mitochondrial Morphology

The morphology of mitochondria was analyzed using the Tomm20 marker. All the images were analyzed with an ImageJ/Fiji macro developed by our group (Šimunić et al., under review).

### 2.11. Statistical Analyses

Statistical analyses were conducted in GraphPad Prism version 8.00 (GraphPad Software, San Diego, CA, USA). All the experiments were repeated 3 times, and their significance was quantified using the *t*-test. A *p* value of *p* < 0.05 was considered statistically significant (*), *p* < 0.01 was considered highly statistically significant (**) and *p* > 0.05 was considered nonsignificant (ns). The data are presented as mean ± SEM (Standard error of mean).

## 3. Results

### 3.1. One and Two Weeks of In Vitro Differentiation of Neural Stem Cells Yields Various Precursors of the Nervous System

The stemness of the cells used for in vitro differentiation was demonstrated with markers Sox2 and nestin. Their presence was clearly shown on the first day of differentiation (D1) in a great majority of the observed cells (Figure 1). To demonstrate the sequence of in vitro differentiation of NSCs, we analyzed the expression of nestin, MAP2, and GFAP in three distinct timepoints: D1, D7, and D14. In cells belonging to the D1 group, the dominant population was nestin-positive, with only a few immature, mildly GFAP- and Map2-positive cells being visible (Figure 2a,c,e; upper row—control group). On the other hand, the cell culture of D7 was primarily composed of GFAP-positive astrocytes, with a smaller number of MAP2-positive neurons (Figure 2a,c,e; upper row—control group). Finally, on D14, the astrocytes were the predominant cell type present within the culture (Figure 2a,c,e; upper row—control group). As all three timepoints exhibited distinct cellular populations, these results indicate that the selected temporal steps of differentiation present a valid model that mirrors various stages of maturity of the nervous system.

### 3.2. The Influence of Oxygen and Glucose Deprivation on Cytotoxicity and Differentiation of Neural Precursors Does Not Depend on the Level of Their Maturity

To quantify the effects and confirm that our model of oxygen–glucose deprivation for 24 h causes cytotoxicity similar to the one observed in perinatal hypoxia–ischemia, we measured the levels of lactate dehydrogenase hydrate (LDH). The leakage of LDH from the cells and into the cultivation medium directly correlates with the level of cell membrane disruption that, ultimately, leads to cell death. On D1, 35% of cells cultivated in the control group underwent cell death. When the same stage of neural precursors was exposed to OGD, the extent of cell death observed reached 75%. On D7, the control group exhibited a slight decrease in the extent of cell death (30%), while the OGD group exhibited the same results observed on D1. A similar trend was also found at D14, wherein the extent of cell death in the control group was slightly decreased compared with the previous stages, while that of the OGD group reached 75% (Figure 3).

To investigate how OGD influences the differentiation of the neural precursors on D1, D7, and D14, we compared the cells in the control and OGD environments using qPCR and immunocytochemistry. This was accomplished through the quantification of mRNA levels of nestin, MAP2, and GFAP as well as descriptive ICC image analysis. During all three timepoints of interest, the expression of nestin decreased after oxygen–glucose deprivation, with significant differences observed on D1 and D7 (Figure 2a, lower row, and qPCR in Figure 2b). The analysis of astrocyte marker GFAP, following OGD treatment, showed an increase during D1, and a subsequent decrease on D7 and D14 (Figure 2c, lower row, and qPCR graphs in Figure 2d). The analyses of neuronal marker Map2 revealed a significant decrease in all the measured timepoints (Figure 2e, lower row, and qPCR graphs in Figure 2f).

### 3.3. The Morphological Parameters of Mitochondria Change during Cell Differentiation Are Influenced by the Lack of Oxygen and Glucose

Since we hypothesized that the lack of oxygen and glucose influences mitochondrial morphology, we used Tomm20, the marker of the outer membrane, to study the mitochondrial network over three distinct timepoints. The measured parameters include the total area of the mitochondria, the length of the mitochondrial filaments, and the number of branching. In addition, similar to the study by Ahmad et al., we calculated the ratios of the following distinct morphological forms of mitochondria: tubular filaments, intermediate filaments, and punctate filaments [18].

The analysis of the total mitochondrial area for each cell, wherein the Tomm20 area was normalized to the cell number, revealed that the total area, length, and the number of the branches of the mitochondrial network increased during NSC maturation. Thus, and in comparison to the early neuronal precursors, a four-fold increase in Tomm20 area and mitochondrial branching was detected. Additionally, a five-fold increase in Tomm20 length was visible after 14 days of differentiation (Figure 4). 

Our next question was how the lack of oxygen and glucose influences the area, branching, and total length of the mitochondrial filaments. Although the OGD treatment caused a decrease in total Tomm20 positive area in all three timepoints, a statistical significance was found only on D1 and D7. Interestingly, along with a decrease in the total mitochondrial surface, we also observed a 25–50% decrease in the total length and branching of the mitochondrial filaments in all three timepoints (Figure 5).

After quantifying the total area of the mitochondria, as well as the total length of their filaments, analyses of distinct forms of mitochondria were performed. Following normalization with the total Tomm20 area, we found that the ratio of tubular filaments increased over time in the control group. At the same time, while the ratio of punctate filaments was shown to be decreased, the levels of intermediate fragments appeared to be constant throughout. These results indicate that mitochondrial fusion cycles become more prominent in later stages of differentiation, having a crucial role in the formation of complex mitochondrial networks. As opposed to the patterns visible within the control group, the number of tubular filaments decreased with the differentiation time in the OGD-treated group, while the number of intermediate and punctate filaments increased (Figure 6). These findings were confirmed after 3D analysis of mitochondrial TMRE where a slight increase in mitochondrial sphericity was detected (data not shown). Accordingly, this suggests that the lack of oxygen and glucose promotes fission cycles during differentiation.

### 3.4. The Lack of Oxygen and Glucose Increases the Number of Oxygen Radicals in the Mitochondria and Causes Their Hyperpolarization

With the goal of quantifying some major parameters of mitochondrial metabolism during the differentiation of neural precursors, especially as they pertain to mitochondrial exposure to the lack of oxygen and glucose, we measured the levels of oxygen radicals and the mitochondrial membrane. The quantification of levels of oxygen radicals that are present in the mitochondria revealed an increase in all three timepoints during which the cells were exposed to OGD, with significant differences in D1 and D14 (Figure 7).

The use of the TMRE to measure the mitochondrial membrane potential allowed us to detect the hyperpolarization of the mitochondrial membrane in all three timepoints, with a statistical significance observed on D1 and D7. No uniform results were observed on D14 (Figure 8).

### 3.5. The Intensity of Autophagy and Mitophagy Depends on the Cells’ Differentiation Stage 

Since we detected a rather uniform increase in the number of oxygen radicals, coupled with the hyperpolarization of the mitochondrial membrane following exposure to OGD in all three developmental stages, we also quantified PINK1 (PTEN-induced kinase 1), a marker of the physiological function of mitochondria linked to mitophagy. In agreement with the results obtained using MitoSOX and TMRE, we found a statistically significant increase in PINK1 on D7 and a non-significant increase on D14 (Figure 9). The number of proteins secreted by cells on D1 was too low to draw any conclusions, so they were omitted from this analysis.

By detecting the mitochondrial outer-membrane protein FUNDC1 (FUN14 Domain Containing 1) in both qPCR and WB, a 3.5-fold decrease in FUNDC1 was observed on D7. This indicates the induction of receptor-mediated mitophagy (Figure 10). On the other hand, even though some increase in FUNDC1 levels were observed on D14, the changes were not significant (Figure 10).

Since clear signs of mitophagy were detected on D7, we also quantified the levels of autophagy. With that in mind, a clear increase in LC3-II, a marker of autophagy, was observed during both D7 and D14. Coupled with a decrease in p62 levels, these findings indicate an ongoing induction of autophagy (Figure 11).

Since an increase in LC3-II might indicate an increase in autophagosome synthesis or a reduction in their removal, we used a Baculovirus containing LC3B-linked GFP protein chimera. This was followed by colocalization with Lysotracker since it specifically labels lysosomes within cells. Here, an increase in LC3/GFP-Lysotracker colocalization in all three specific timepoints was observed, with a significant difference in D7 and D14 (Figure 12).

## 4. Discussion

The deregulation of circulation, which leads to the lack of oxygen and glucose within the tissue, is the backbone of many diseases that cause life-long impairment, including stroke and perinatal brain damage. Although perinatal brain damage can be separated into five distinct entities, namely hypoxic–ischemic encephalopathy, intraventricular hemorrhage, periventricular leukomalacia, and perinatal stroke, they have one pathophysiological element in common: oxygen and glucose deprivation. With one-third of neonates affected by perinatal brain damage, those who do not die develop severe seizures, and motor, cognitive, and memory impairments, alongside cerebral palsy. Since this represents approximately 30–40% of survivors, it poses a great medical burden against which our therapeutic options are still insufficient and, therefore, lacking [19]. With the goal to elucidate how the sudden lack of oxygen and glucose influences the immature cells of the nervous system, we exposed an in vitro model of neural precursors to OGD during three stages of their development to OGD. The first step was to fine-tune the levels of OGD exposure needed to trigger measurable cell death with an LDH test, including all the major markers of differentiation present. This was followed by a direct comparison of their capability for differentiation. With the exception of the earliest precursors, which expressed more GFAP, the precursors exposed to OGD exhibited decreased nestin, MAP2, and GFAP gene expression in all differentiation stages. 

Considering that one of the goals of this research was to quantify the influence of oxygen–glucose deprivation on the mitochondria and mitochondria-related processes such as mitophagy, we simultaneously developed a protocol for semi-automatic quantification of the parameters needed in such an investigation, namely Lusca. Lusca is an ImageJ/Fiji macro designed for detailed and accurate recognition of circular, fibrillar, and punctate structures in digital images of tissue or cells (Šimunić et al., under review). This macro allowed us to quantify the total area, the number of mitochondrial branches, and the total length of mitochondrial filaments in developing cells of the nervous system using Tomm20, a marker of the mitochondrial membrane. Interestingly, our results demonstrate that the cells significantly increased the ratio of the cytoplasm volume normalized to the total cell volume filled with mitochondria through the increasing stages of differentiation. One potential reason behind this is the fact that, as cells mature, the mitochondria gradually become the more important organelles in cells of the nervous tissue. When exposed to OGD, the cells within all three differentiation timepoints reacted by decreasing the total volume, length, and branching of the mitochondrial membranes. This effect was most prominent in cells on D7.

In order to quantify these changes in mitochondrial parameters and morphology, we used Lusca to separate the observed mitochondria into three distinct forms—tubular, intermediate, and punctate—with tubular mitochondria denoting the healthy and punctate denoting damaged mitochondria undergoing fission [18,20]. Since the mitochondrial network and its metabolism are a very dynamic system, undergoing constant fusion/fission cycles, the balance between the main fusion proteins Opa1 (optic atrophy 1) and Mitofusin 1 and 2, and the fission protein Drp1 (dynamin-related protein), is an important mechanism underlying the regulation of mitochondrial architecture [21]. In an undifferentiated state, immature cells, such as NSCs, maintain a low metabolic activity characterized by a low oxygen consumption rate, fragmentation, and glycolysis [22,23]. This hypothesis is supported by the results of our study, wherein, during the first day of differentiation, almost exclusively SOX2- and Nestin-positive cells were observed, characterized by a small number of Tomm20-positive regions. Additionally, cell differentiation is also initiated and, therefore, regulated through changes in ATP production, namely its increase. This is accomplished by an increase in the mitochondrial membrane potential and changes in the electron transport chain located in the inner mitochondrial membrane [23]. Similar phenomena can also be seen within our study, wherein the growth of the mitochondrial network was observed on D7 and D14, depicted as an increase in the Tomm20-positive area and filament length.

Since appropriate mitochondrial functioning is a crucial factor in determining the proper development of the nervous system, our research also included investigations into the reaction of neural precursors to OGD. As expected, these effects depend on the cells’ developmental stage. Interestingly, significant changes in the cell mitochondria, namely an increased incidence of punctate elements, were observed only on D7. Based on these observations, it appears that cells in the middle stage of differentiation react to OGD with a significant reduction in the tubular form of mitochondria, which are also prone to mitochondrial fission. These findings are in accordance with other research within the field, wherein evidence of mitochondrial fragmentation following ischemia, including an increase in mitophagy, was observed [16]. Alongside fragmentation, we noticed that the affected mitochondria are more spherical, which possibly correlates with its swollen and rounded ultrastructural appearance analyzed using electron microscopy [21]. Such an increase in the ratio of punctate mitochondria could be a mechanism through which cells accelerate the removal of damaged mitochondria and improve their chances of survival [24].

As the polarization of the mitochondrial membrane plays a significant role in energy transformation during oxidation, and therefore ATP production, we also evaluated ROS production and the corresponding MMP. Any imbalance in the oxidative status of the system results in misfolding of proteins, lipid peroxidation, genome instability, general cellular toxicity, and opening of mPTPs (mitochondrial permeability transition pores) [20,25]. Even though an increase in ROS production in all three stages of differentiation following OGD was detected, no significant difference was observed on D7. This can possibly be explained by the fact that the largest number of neurons are present on D7 under normal conditions. Once they are exposed to OGD, they die. The fact that neurons produce more oxygen radicals than astrocytes explains why our results depicted a lower presence of ROS on D7 [26].

Similarly, TMRE revealed a hyperpolarization of the mitochondrial membrane in all three stages of cell differentiation. This hyperpolarization may be explained as a consequence of cellular shrinkage and subsequent mitochondrial accumulation. Such findings are supported by those of Agarwal et al., who suggested that cellular heat-shock response—usually induced by an increase in temperature, ischemia, ROS production, and other stressors—leads to perinuclear mitochondrial clustering and an increase in overall mitochondrial intensity [27]. Alternatively, since we also detected excessive ROS production, the hyperpolarization of the mitochondrial membrane could be due to the phenomenon called RIRR (ROS-induced ROS release) that opens mPTP as a part of the cellular adaptive mechanism of releasing ROS to the environment [20]. The research by Ward et al. suggested that delayed apoptotic injury of neurons is first characterized by a depolarization period, followed by the hyperpolarization of the mitochondrial membrane [28]. Under normal conditions, ROS increase is accompanied by a decrease in the mitochondrial membrane potential. On the other hand, Zorov et al. reported that, under certain conditions, ROS overproduction could also lead to the occurrence of a hyperpolarization period [20]. Similarly, Korenić et al. described the same phenomenon in astrocytes exposed to OGD [29]. The fact that our cell culture boasts predominant numbers of astrocytes over neurons, as is the case in the human brain, might explain why our results depict clear cases of hyperpolarization of the mitochondrial membrane, as opposed to depolarization noted in experiments conducted on predominantly neuronal cultures [30].

Further analyses performed on markers of mitophagy, namely PINK1 and FUNDC1, suggest that cells already undergo this process on D7 and D14. PINK1 is a mitochondrial serine–threonine protein kinase that acts as an important mitochondrial quality control system. In physiological conditions, it is effectively imported to the inner mitochondrial membrane, where it is subsequently cleaved and degraded. On the other hand, PINK1 is also known to accumulate on the outer membrane of damaged mitochondria, where it can induce the ubiquitin-dependent removal of mitochondria [31]. Similar to what was observed with PINK1, a significant decrease in FUNDC1, a novel mitochondrial receptor, was observed on D7. This mirrors the results of Liu et al., reporting the hypoxia-induced degradation of mitochondria [32]. To quantify the extent of autophagy, we also evaluated the levels of LC3-II, in combination with its colocalization with a lysosome. The results revealed that the levels of LCR-II were significantly increased during both the middle and late stages of differentiation.

## 5. Conclusions

In conclusion, our study revealed that precursors of the nervous tissue exhibit different and distinct reactions to a sudden lack of oxygen and glucose. Apart from cell death and the loss of the total number of mitochondria, which seems to be a common event, cells in the middle stage of development (which is D7 in our system) were more prone to mitochondrial fragmentation, with clear signs of both autophagy and mitophagy being present. As such, the results of this study facilitate a better understanding of the events occurring in the immature nervous system. Thus, they point towards new directions in designing therapeutic strategies for treating perinatal brain damage.

## Figures and Tables

**Figure 1 brainsci-13-00910-f001:**
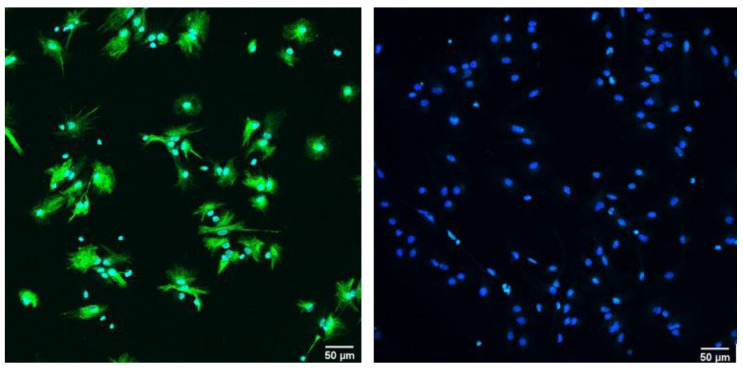
Expression of NSC’s stem cells markers nestin (green) and SOX2 (blue) during the first day of differentiation under 20x magnification, with the scale bar = 50 µm (counterstain with DAPI).

**Figure 2 brainsci-13-00910-f002:**
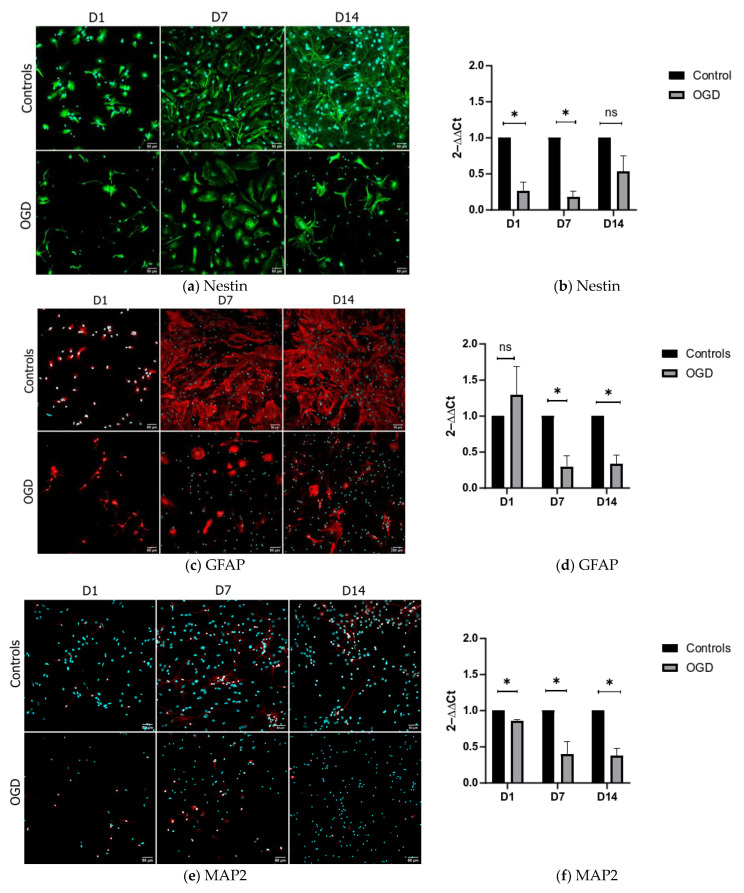
NSC’s differentiation pattern was analyzed at 3 different timepoints using antibodies against nestin (shown in green), GFAP (shown in red (**c**)) and MAP2 (shown in red (**e**)). The upper rows in (**a**,**c**,**e**) depict representative images of control groups, while the lower rows in (**a**,**c**,**e**) depict the cells after exposure to OGD. ICC images were taken on 20× magnification, with the scale bar = 50 µm (counterstain with DAPI (shown in blue)). qPCR results for the same markers, depicted in (**b**,**d**,**f**), are shown as fold differences.

**Figure 3 brainsci-13-00910-f003:**
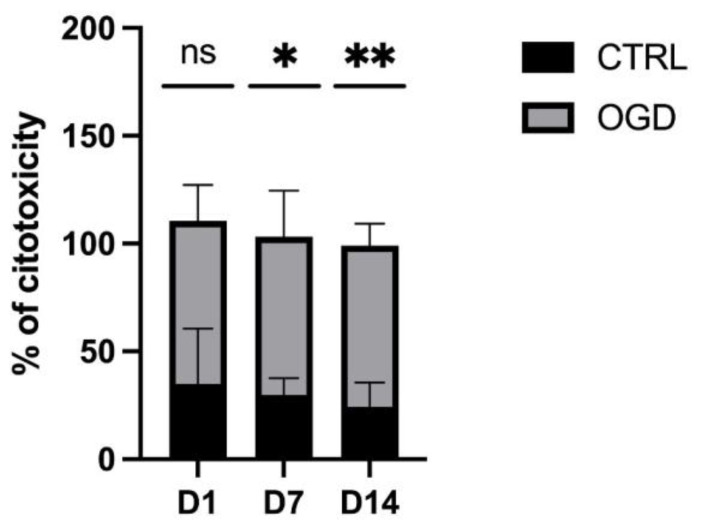
The difference in the percentage of cell cytotoxicity throughout 3 timepoints between CTRL (Control) and OGD-treated groups.

**Figure 4 brainsci-13-00910-f004:**
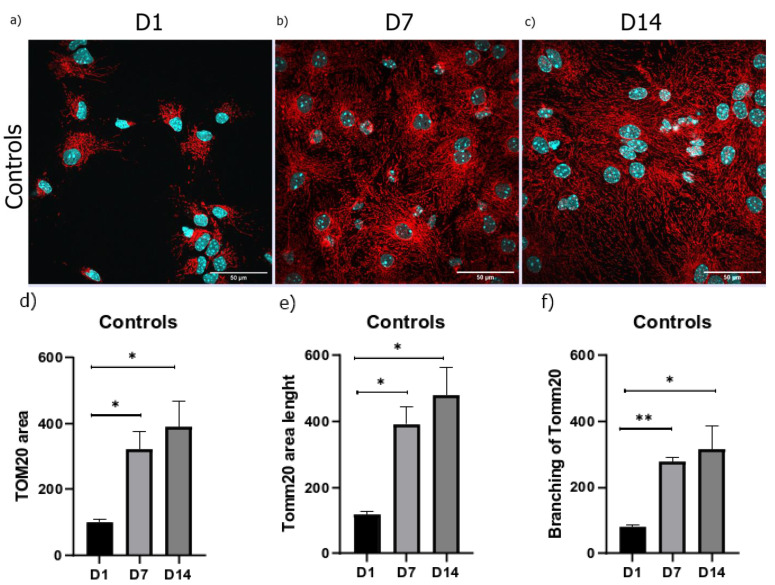
Immunocytochemical staining of (**a**) D1, (**b**) D7, and (**c**) D14 revealed an increase in the number of Tomm20 filaments (shown in red) during cell differentiation under 60× magnification (counterstain with DAPI (shown in blue). The increase in (**d**) area, (**e**) length, and (**f**) branching of the mitochondrial network during cell differentiation. The results are shown as mean ± SEM.

**Figure 5 brainsci-13-00910-f005:**
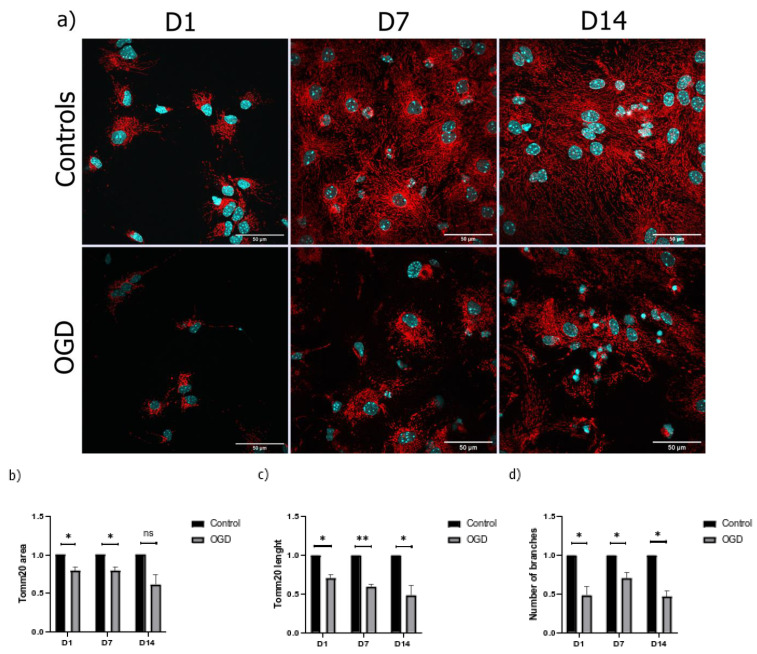
(**a**) Representative immunocytochemical staining of Tomm20 (shown in red) following OGD treatment under 60× magnification (counterstain with DAPI (shown in blue)). Fold difference in (**b**) area, (**c**) length, and (**d**) branching of mitochondrial filaments. The results are shown as a fold difference ± SEM.

**Figure 6 brainsci-13-00910-f006:**
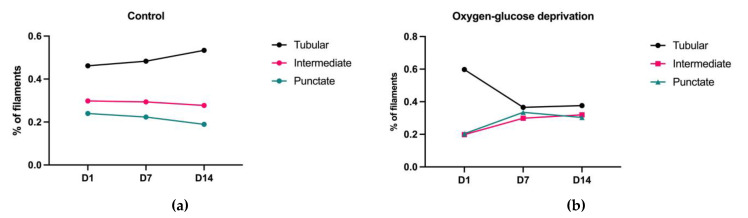
Changes in the ratio of specific mitochondrial subclasses of differentiating NSCs in (**a**) the control and (**b**) OGD groups. The fold difference ± SEM in the percentage of the area of (**c**) tubular, (**d**) intermediate, and (**e**) punctate mitochondria.

**Figure 7 brainsci-13-00910-f007:**
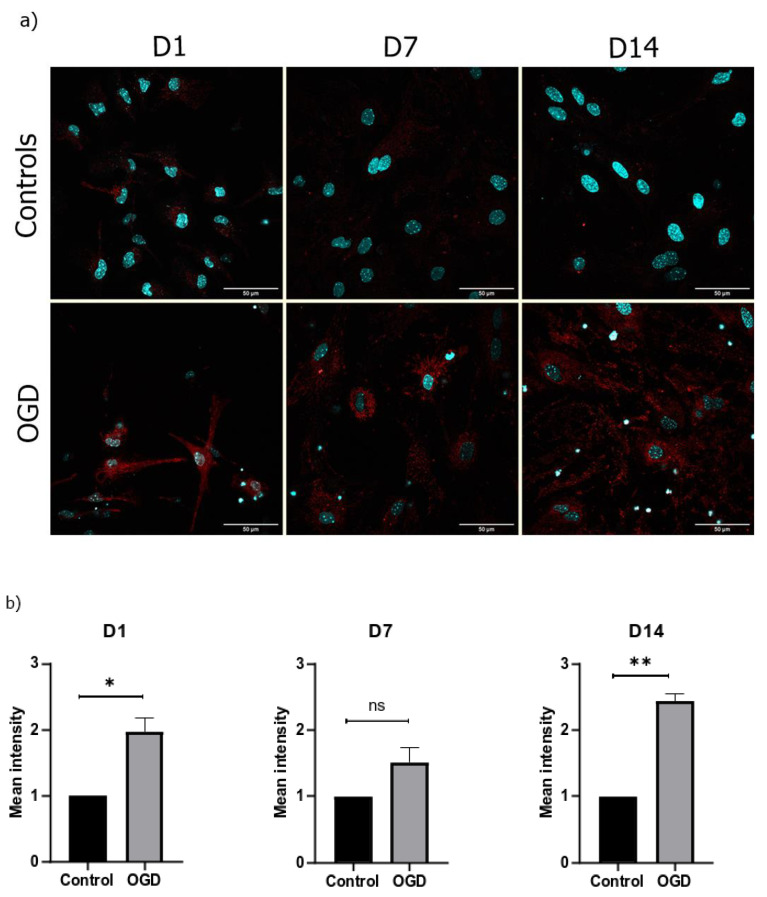
(**a**) Representative pictures of oxygen radicals (shown in red) measured using a MitoSOX kit in the control and OGD-treated groups under 60× magnification (counterstain with DAPI (shown in blue)); (**b**) quantification of the mean intensity on D1, D7, and D14, with the results shown as a fold difference ± SEM.

**Figure 8 brainsci-13-00910-f008:**
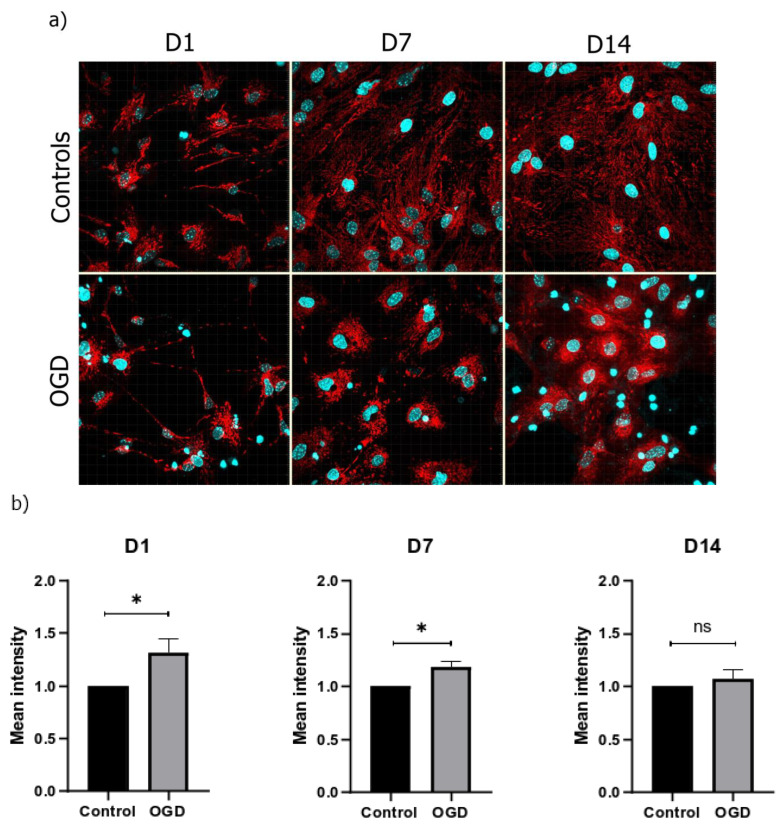
(**a**) Representative pictures of mitochondrial membrane potential (shown in red) in the control and OGD-treated groups under 60× magnification (counterstain with DAPI (shown in blue)); (**b**) mean intensity for D1, D7, and D14 shown as a fold difference ± SEM.

**Figure 9 brainsci-13-00910-f009:**
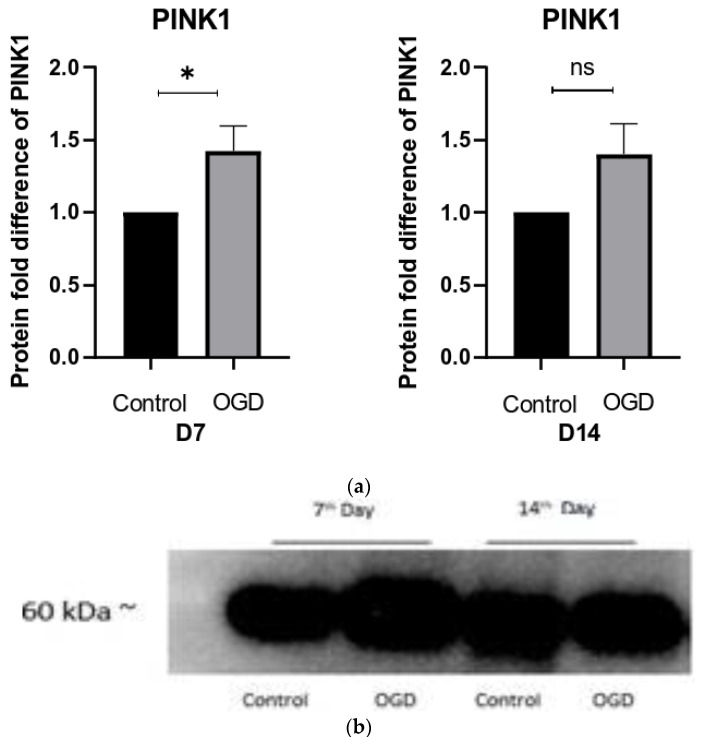
(**a**) The difference in PINK1 expression in control and OGD-treated groups is shown as a fold difference ± SEM; (**b**) the relative band intensities of PINK1.

**Figure 10 brainsci-13-00910-f010:**
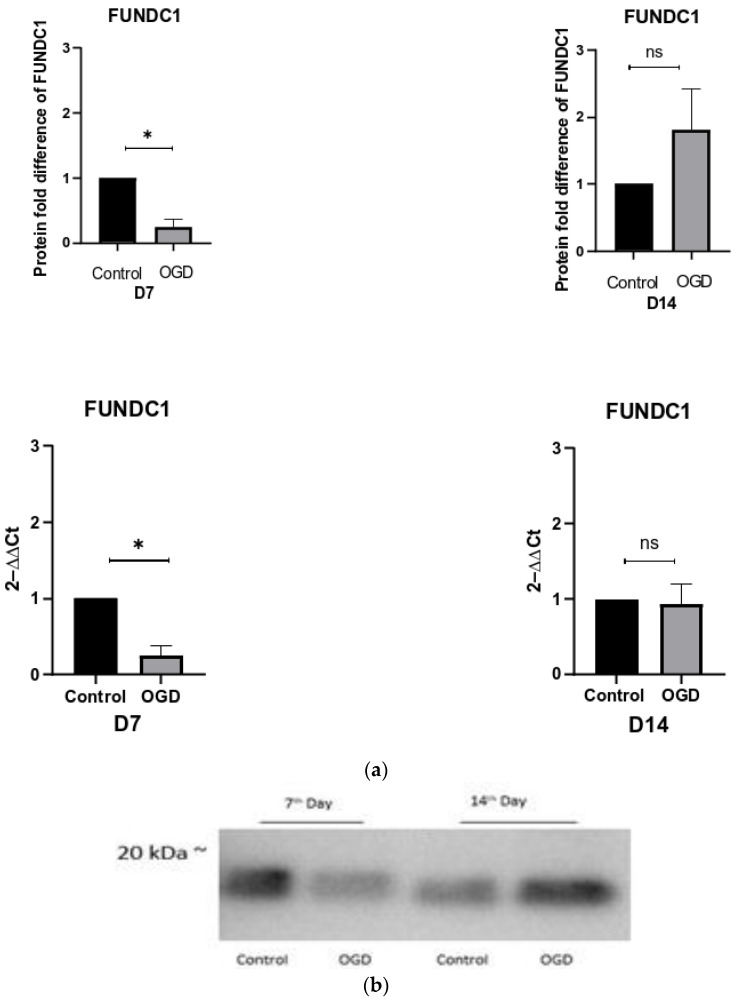
(**a**) The change in protein and mRNA expression of FUNDC1 in control and OGD-treated groups is shown as a fold difference ± SEM; (**b**) the relative band intensities of FUNDC1.

**Figure 11 brainsci-13-00910-f011:**
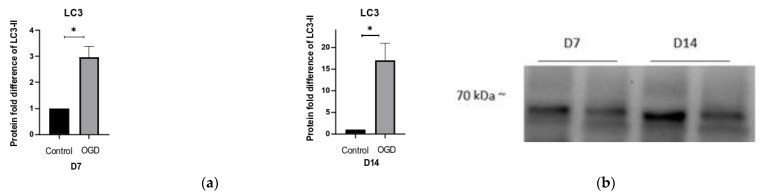
The difference in (**a**) Lc3-II and (**c**) p62 protein expression in control and OGD-treated groups is shown as a fold difference ± SEM; the relative band intensities of (**b**) LC3 and (**d**) p62.

**Figure 12 brainsci-13-00910-f012:**
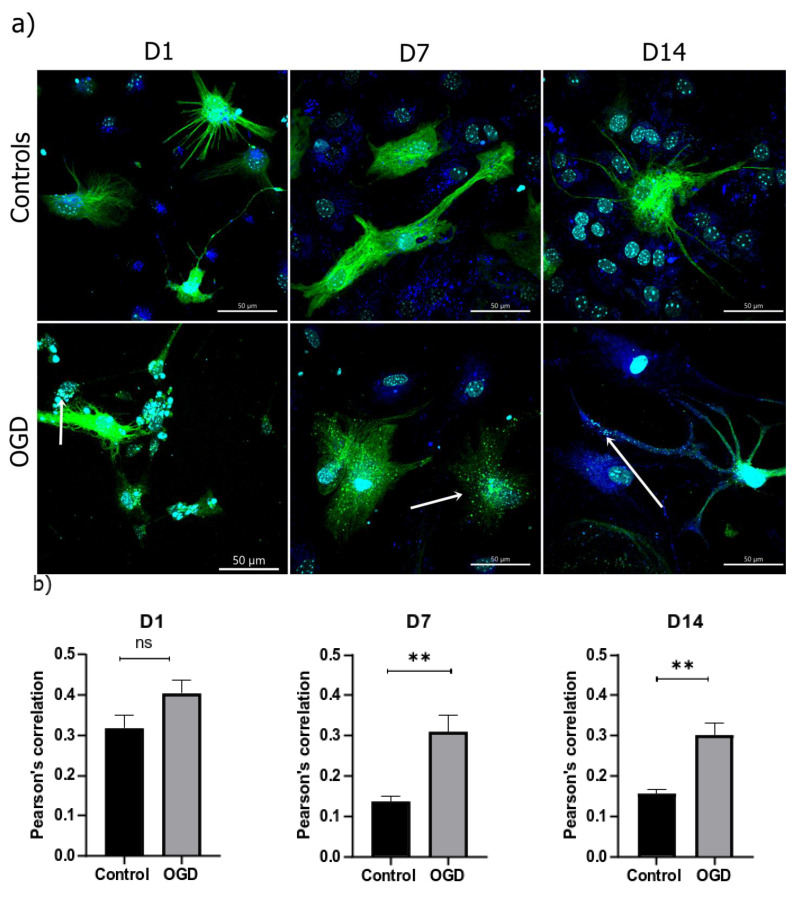
Autophagy analysis was performed by measuring (**a**) the colocalization between Lysotracker (blue) and LC3-GFP (green), in the control group ((**a**), upper row) and following OGD treatment ((**a**), lower row) (counterstain with DAPI); (**b**) Pearson’s correlation coefficient was calculated and is shown as mean ± SEM.

## Data Availability

All data are available upon reasonable request.

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
