# Peer review of "The Oxygen and Glucose Deprivation of Immature Cells of the Nervous System Exerts Distinct Effects on Mitochondria, Mitophagy, and Autophagy, Depending on the Cells’ Differentiation Stage"

_brainsci, 2023, doi:10.3390/brainsci13060910_

Round 1
Reviewer 1 Report
In the manuscript entitled “Oxygen and glucose deprivation of immature cells of the nervous system exerts distinct effects on mitochondria, mitophagy and autophagy depending on the cells differentiation stage“ Jagečić et al. used in vitro neuroprogenitor-based assay to demonstrate that transient oxygen and glucose deprivation (OGD) affect neurogenic differentiation and induce cell death. Unexpectedly, they found that only neuroprogenitors at day 7, and not at days 1 and 14, of differentiation respond to OGD by mitochondrial fission and mitophagy. The author’s intention is, with their in vitro method, to recapitulate the in vivo studies of the effect of OGD in different stages of neurodevelopment. This well-written, interesting manuscript with sound data demonstrates the apparent impact of OGD on neuroprogenitors’s fate.
The manuscript can increase its merit if:
- In Fig. 2, the qPCR results are presented as a fold difference to actin, so the gene expression (nestin, GFAP, MAP2) at different time points can be compared. The 3-time points in the same graph (panel) will probably be better. It is not clear when analyses were done. Is it immediately after 24 of OGD?
- Fig. 5 and Fig. 6 should be supported with representative images. Also, having the 3-time points in one graph without normalization to the control is a better presentation.
- Western blot analyses should be normalized to some internal control, not just by the amount of loaded proteins.
Author Response
Dear Reviewer,
Thank you for reviewing our manuscript, and for all the valuable comments. We agree with the comments, and we have addressed them below. Changes have been made to reflect this, and the revised manuscript with highlighted changes has been attached.
- We thank reviewer for the suggestion regarding Fig. 2. We changed the graph accordingly.
- We added a sentence in the section “Oxygen glucose deprivation model“ that reads (lines 104 - 106):
After this, the cells were placed in a low oxygen (1% O2) incubator for 24 hours, after which the samples were taken. The samples from the control group were collected after the same period, albeit from the cells that were placed in a separate incubator and were growing under normal conditions.
- We added representative immunocytochemistry staining in Fig 5. We also adjusted the graph as per the suggestion., such that all 3-timepoints are shown in one graph. In addition, we also improved Fig 6. by using the violin plot to demonstrate the distribution of data.
- As far as normalization of wb analyses to internal control goes, the protocol which applies housekeeping proteins assumes that their expression does not change in different conditions. However, by looking at our stain-free membrane we have found that the expression of some proteins has changed (usually by being downregulated) following OGD treatment. For this reason, we decided to perform data analysis and quantification with respect to the total amount of proteins, as suggested by articles in the field, and as routinely used in many publications (see PMID: 23085117).

Reviewer 2 Report
The article by Jagečić et al is devoted to the study of the effect of oxygen and glucose deprivation on the structural and functional features of neural cell precursors mitochondria during three stages of their differentiation: day 1, day 7 and day 14. The authors demonstrated cell survival, changes in the size of mitochondria and their functional activity, and also determined the processes of mitophagy and autophagy under these conditions. The authors state that their findings could play a role in guiding the future developments of novel therapeutic approaches targeting perinatal brain damage during specific stages of nervous system development.
Comments:
1. The main remark is that the purpose of this work is not very clear. What do the authors want to show by creating oxygen and glucose deprivation in neural cell precursors at various stages of their differentiation? Why are these 3 stages important? To what extent do the results correlate with what can happen with ischemic damage to the nervous system? It is necessary to understand the novelty of this study in more detail.
2. The experiment with the determination of cytotoxicity is not clear. What induces cytotoxicity? Why is cytotoxicity about 30% in the control group?
3. The authors write that in the case of oxygen and glucose deprivation, mitochondrial hyperpolarization and, at the same time, mitophagy are observed. However, according to numerous literature data, mitochondrial depolarization is one of the stimuli of mitophagy. How can the authors comment on their results? In addition, the authors mention mitochondrial permeability transition pores as a mechanism for increasing the production of reactive oxygen species. However, the formation of mitochondrial pores cannot be accompanied by mitochondrial hyperpolarization. It is also necessary to understand why mitochondrial hyperpolarization and strong cytotoxicity are observed simultaneously.
4. Authors must provide scans of all WBs in supplementary materials.
Author Response
Dear Reviewer,
Thank you for reviewing our manuscript, and for all the valuable comments. We agree with the comments, and we have addressed them below. Changes have been made to reflect this, and the revised manuscript with highlighted changes has been attached.
- Pertaining the purpose of our work not being very clear, our “Introduction” section reads as follows (lines 65 - 70):
Furthermore, even though many publications reported the occurrence of autophagy and mitophagy in the nervous system during various stages of its development, including in the neonatal neurons, a direct comparison of cells throughout these stages of differentiation is lacking [17]. With that, the main goal of this work was to use our expertise in neural cell precursors and investigate the influence of oxygen and glucose deprivation on their early, mid, and mature stages of differentiation.
Furthermore, our “Discussion” section reads (lines 468 - 480):
Deregulation of circulation, which leads to lack of oxygen and glucose within the tissue, is in the backbone of many diseases that cause life-long impairment – including stroke and perinatal brain damage. Although perinatal brain damage can be separated into five distinct entities, namely hypoxic-ischemic encephalopathy, intraventricular hemorrhage, periventricular leukomalacia, and perinatal stroke, they have one pathophysiological element in common – oxygen and glucose deprivation. With one third of neonates affected with perinatal brain damage, those who do not die develop severe seizures, motor, cognitive and memory impairments, alongside cerebral palsy. Since this represent approximately 30–40% of survivors, it poses a great medical burden against which our therapeutic options are still insufficient and, therefore, lacking [19]. With the goal to elucidate how the sudden lack of oxygen and glucose influences the immature cells of the nervous system, we exposed an in vitro model of neural precursors to OGD during three stages of their development to OGD.
So, to answer the question directly, oxygen and glucose deprivation is a very well-known in vitro model of stroke. Interestingly, while there are many publications focused on stroke in adults, data about events occurring in immature neural tissue, i.e. the tissue affected by perinatal stroke is lacking. So, creating OGD in neural precursors is a model of perinatal stroke. Since cells in immature brain are present in various stages of immaturity/maturity, we have chosen three stages of in vitro differentiation. These time points were chosen primarily based on our huge experience with neural stem cells. Of course, it is impossible to precisely correlate each of these steps to every cell in the immature human brain, but, like every model, we believe that it corresponds well to at least some stages of in vivo development.
- We thank the reviewer for his insight. To answer the question, cytotoxicity is a well-known effect of sudden lack of oxygen and glucose. It stands in the center of tissue damage occurring during stroke (see https://www.mdpi.com/1422-0067/24/8/7106). As such, cell death reaching around 30% is not surprising. Such cell cultures are composed of many cell types and, to some extent, cell death is always present. In many publications, researchers normalize cell death to control conditions, i.e. they are not revealing the cell death in the control conditions. However, in those articles which use presentations of results like we do, it is visible that cell death is a normal element also present under control conditions (see PMID: 33621193 and 23001052). We hope that answers the question.
- With respect to mitochondrial hyperpolarization, we agree that hyperpolarization of the membrane was not an expected finding, but that is the reason why we interpreted and analyzed this finding in as many details as possible in the “Discussion” section (lines 547 - 566):
Similarly, TMRE revealed a hyperpolarization of the mitochondrial membrane in all three stages of cell differentiation. This hyperpolarization may be explained as a consequence of cellular shrinkage and subsequent mitochondrial accumulation. Such findings are supported by Agarwal et al who suggested that cellular heat shock response – usually induced by an increase in temperature, ischemia, ROS production and other stressors – leads to perinuclear mitochondrial clustering and an increase in overall mitochondrial intensity [27]. Alternatively, since we also detected excessive ROS production, the hyperpolarization of the mitochondrial membrane could be due to the phenomenon called RIRR (ROS induced ROS released) that opens mPTP as a part of the cellular adaptive mechanism of releasing ROS to the environment [20]. The research by Ward et al. suggested that delayed apoptotic injury of neurons is first characterized by a depolarization period, followed by hyperpolarization of the mitochondrial membrane [28]. In normal conditions, ROS increase is accompanied by a decrease in the mitochondrial membrane potential. On the other hand, Zorov et al. report that, under certain conditions, ROS overproduction could also lead to the occurrence of a hyperpolarization period [20]. Similarly, Korenić et al. describe the same phenomenon in astrocytes exposed to OGD [29]. Since our cell culture boasts predominant numbers of astrocytes over neurons, as is the case in the human brain, this might explain why our results depict clear cases of hyperpolarization of the mitochondrial membrane, as opposed to depolarization noted in experiments done on predominantly neuronal cultures [30]. We added one new reference which said that cellular heat shock response (induced by increased temperature, ischemia, ROS production and other physical and chemical stressors) leads to mitochondrial accumulation with increase in overall mitochondrial intensity [27].
- Scans of all WBs have been included in the supplementary materials.
